

# The role of awareness of repetition during the development of automaticity in a dot-counting task

Craig P. Speelman and Emma Shadbolt

School of Arts and Humanities, Edith Cowan University, Joondalup, Western Australia, Australia

## ABSTRACT

This study examined whether being aware of the repetition of stimuli in a simple numerosity task could aid the development of automaticity. The numerosity task used in this study was a simple counting task. Thirty-four participants were divided into two groups. One group was instructed that the stimuli would repeat many times throughout the experiment. The results showed no significant differences in the way automatic processing developed between the groups. Similarly, there was no correlation between the point at which automatic processing developed and the point at which participants felt they benefitted from the repetition of stimuli. These results suggest that extra-trial features of a task may have no effect on the development of automaticity, a finding consistent with the instance theory of automatisation.

## INTRODUCTION

It is commonly observed that repeatedly practicing a task can lead to a change in performance over time from slow and deliberate to fast and seemingly without thought (*Epstein & Lovitts, 1985*; *Logan, 1988*; *Wilkins & Rawson, 2011*). In Psychology, these two kinds of cognitive processing are often referred to as controlled and automatic processing. Controlled processing is said to occur where performance is deliberate, limited by memory capacity and requires attention (*Schneider & Shiffrin, 1977*). Automatic processing occurs without a person's control, without capacity limits, without necessarily demanding attention and develops over time with much practice of a task (*Epstein & Lovitts, 1985*; *Logan, 1988*; *Logan, 1990*). With practice, a person can transition from controlled to automatic processing. Evidence for the transition from controlled to automatic processing after practice has been found in many simple cognitive tasks including alphabet-arithmetic, lexical decision, Stroop paradigms, relative judgement, categorization, visual search, and dual-task scenarios (*Augustinova, Flaudias & Ferrand, 2010*; *Hélie, Waldschmidt & Ashby, 2010*; *Hommel & Eglau, 2002*; *Loft, Humphreys & Neal, 2004*; *Logan, 1990*; *Logan & Klapp, 1991*; *Shiffrin & Schneider, 1984*).

*Schneider & Shiffrin (1977)* reviewed and summarised research in detection, search and attention studies relating to automaticity. They concluded that controlled processing is a temporary activation of a new mental sequence allowing performance of a specific task

Corresponding author
Craig P. Speelman,
c.speelman@ecu.edu.au

that is not yet learned. Due to being a new task, the mental sequence required to respond is relatively easy to modify and use in new situations. This controlled processing also requires short-term memory capacity and attention in order for a correct response to be made. In contrast, automatic sequences are well established and do not require attention as the connection between a stimulus and response has been consistently mapped many times. This allows a response to occur regardless of the memory load required as the whole sequence is automatically activated when the stimulus is presented. As a result, automatic processing is not constrained by short-term memory capacity limits.

As already mentioned, the development of automaticity has been demonstrated in many simple tasks; all of which tend to be consistent in nature, requiring a stimulus to be mapped directly and consistently to a response over a period of practice (*Strayer & Kramer, 1990*). Given this environmental consistency, memory for responses can be utilised automatically rather than generating a response in a controlled manner for each stimulus. That is, rather than generating a response by working through several processing steps, a stimulus can automatically activate a memory for the required answer. *Logan (1988)* demonstrated that a memory-based account provides a credible explanation of the phenomenon of automaticity.

*Logan (1990)* reported results of several experiments that provide significant support for his instance theory of automaticity. Instance theory proposes that the transition from controlled to automatic processing reflects a race between the application of an algorithm to produce a response and a memory process that retrieves a response based on past experiences (*Logan, 1988*; *Wilkins & Rawson, 2011*). For example, when solving $4 \times 6 =?$ we might move from generating the answer through an addition strategy (i.e., $6 + 6 + 6 + 6$) to directly remembering an answer (i.e., 24). The state of automaticity is said to have been attained when retrieval of an answer is faster than the calculation of an answer. The theory states that during repeated performance of a task, mental representations of the task, the response and the outcome accumulate and are stored in memory. These mental representations are referred to as instances. These instances are also retrieved during performance. Initially this retrieval is slower than the generation of an answer (*Choplin & Logan, 2005*; *Logan, 1988*; *Logan, 1990*). As experience grows, more instances accumulate in memory. This increases the chance that retrieval of an instance can occur faster than the generation of an answer, and an automatic response (i.e., retrieving and responding on the basis of an instance) becomes more likely (*Logan, 1990*).

One clear example of how the development of automaticity fits the explanation provided by the instance theory was provided by *Lassaline & Logan (1993)*, using a simple dot counting task. Four participants were presented with images of dots, ranging in number from six to eleven, in a random arrangement on a computer screen. The task involved counting and indicating the number of dots on the screen as quickly and accurately as possible. The dot patterns were repeated for four blocks of 120 trials. Each participant completed 13 sessions of the four blocks, totalling 5,760 trials. Patterns were presented in a random order within blocks.

*Lassaline & Logan*'s *(1993)* results with the dot counting task provide a clear example of the transition from controlled to automatic processing with practice. Early in practice,
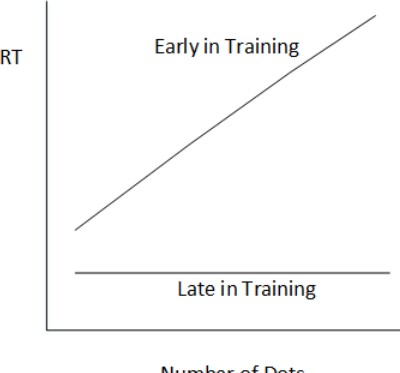

**Figure 1** Reaction time in the dot counting task as a function of number of dots in each stimulus picture.

participants had no choice but to count the dots in each image in order to make a response. At this point the counting of dots is a controlled process requiring attention. According to the instance theory, participants were applying an algorithm (counting) for generating a response. That participants are engaging in such a controlled form of processing is indicated by the fact that their reaction times are directly related to the number of dots in an image (see Fig. 1). After a period of practice, RT is no longer related to the number of dots in an image—participants responded just as fast to an image with six dots as one with eleven dots (Fig. 1). The instance theory explanation for this result is that participants were now recognising the dot patterns and remembering the correct answer, rather than generating the answer by counting. Thus participants had developed automaticity. In Lassaline and Logan's experiments, the slope of the line relating RT and the number of dots in an image in the first session was significantly greater than that for all other sessions, with the slope values reaching asymptote by session four reflecting when automaticity was reached.

Although the existence of automatic processing has been largely supported in previous research there are few clear findings with regard to the influence of the level of awareness of, or attention given to important characteristics of a task (*Boronat & Logan, 1997*; *Epstein & Lovitts, 1985*). It is evident that characteristics of the stimuli attended to during practice, such as spatial extent, item identity and pattern are important in the memory retrieval process (*Green, 1997*; *Kramer, Di Bono & Zorzi, 2011*). However what is unclear in the literature is whether deliberately directing attention to such characteristics of the stimuli before practice commences can impact the rate at which automaticity is reached. In the instance theory, the importance of these characteristics and their preservation in the instance representations are determined by the nature of the task and an attentional filter (*Lassaline & Logan, 1993*). An attentional filter determines characteristics of a task that are noticed by the performer, based upon the importance of these characteristics for completing the task. The more important a characteristic is for completing a task, the more likely it will be attended to, and preserved in memory as instances (*Lassaline & Logan, 1993*; *Moors & De Houwer, 2006*). Automaticity is highly task specific, developing in consistent

task environments where the nature of the task does not change over practice (*Palmeri, 1997*). Similarly, instances are highly task specific representations in memory of the way in which stimuli and responses are associated. Attention is important in the early stages of practice to learn the specific way in which a task needs to be performed (*Palmeri, 1997*; *Yamaguchi & Proctor, 2011*). According to instance theory, attention is required while developing automatic processing of the task, but after automaticity is reached, attention is less important. Given this, it follows that by deliberately directing attention to the nature of the association between stimuli and responses before the development of automaticity, the rate at which automatic processing develops may be accelerated.

The role of attention in learning has also been examined in the context of implicit sequence learning. For example, *Wilkinson & Shanks (2004)* reported results that suggested that what participants learned about a sequence was affected by explicit instructions designed to change the focus of their attention during the task. Similar results were reported by *Jiménez & Méndez (1999)*, who concluded that participants can only learn to associate those items that are concurrently the focus of attention, and so drawing attention to the predictive relationships between items can facilitate learning.

The purpose of this study was to test the influence of providing important information about the nature of a task on the rate of automaticity development. A dot counting task similar to the one used in *Lassaline & Logan*'s *(1993)* study was used as it provided a sound example of how automaticity develops in a simple task. The task allowed the manipulation of awareness of the repetition of dot patterns before the task via pre-experimental instructions. The first aim of the present study was to test whether bringing awareness to the repetition of patterns could encourage memory retrieval and increase the speed at which automatic processing developed. ''The instance theory says that instances are … separate representations of episodic co-occurrences—and that attention determines which co-occurrences go into an instance'' (*Logan & Etherton, 1994*, p. 1046). Thus, drawing attention to the repetitive nature of the dot stimuli was predicted to encourage participants to pay attention to whatever might discriminate between stimuli in a way that could trigger a memory for the correct answer associated with each stimulus, and so avoid having to count dots. For instance, it is possible that awareness of repetition would indicate to participants that they only needed to discover the co-occurrence between some of the dots in each pattern and the correct solution, and so only pay attention to some of the dots. If this were the case, a faster transition to automaticity would be expected. For those participants who were not informed of the repetitive nature of the stimuli, they would need to discover these co-occurrences after they had noticed the repetition of stimuli for themselves.

The second aim of this study was to assess whether participants who were instructed to pay attention to the repetition of the pictures felt that this helped them perform the task. The participants' knowledge of when automaticity developed for them was examined with a post-experimental interview. This enabled the assessment of whether the time at which participants reported reaching automaticity was correlated to when the data suggested

Practice Stimuli

6 dots          7 dots          8 dots          9 dots

**Figure 2  Practice stimuli used in the initial four practice trials.**

that they had attained it. This also established whether pre-experimental awareness of the repetition of items led to greater awareness of the process of developing automaticity.

# MATERIALS & METHODS

## Design

This experiment had a two-group design, with one experimental group and one control group. The group variable was prior awareness of the repetition of dot patterns, which was manipulated through the administration of written instructions prior to commencement of the task. Reaction time on each trial was used to determine when someone had attained automaticity, and to compare when this occurred for the control group and the experimental (aware) group. A post-test interview was also conducted with each participant. The interview consisted of questions relating to the participants' awareness of when automaticity developed over the period of practice.

## Participants

Thirty-five participants were tested, with one participant's results excluded from analysis as a result of low accuracy (less than 70% over all experimental trials). The two groups consisted of 17 participants each. The participants ranged in age from 18 to 62 years. There were 17 males and 17 females. The final sample size of 34 participants was considered adequate, as this sample size exceeds the four participants used in *Lassaline & Logan*'s *(1993)* study which clearly demonstrated automaticity and yielded clear, statistically significant effects. Participants were randomly assigned to the groups.

## Materials

The stimuli consisted of four practice dot pictures and six experimental pictures. The four practice stimuli were in a clear symmetrical, domino-like formation (see Fig. 2). These stimuli were designed to be easily recognised patterns for the participants to count for the numbers 6–9. Conversely the experimental dot pictures were designed to look random in order to present patterns that are not commonly encountered in everyday life. The use of novel stimuli encourages algorithmic (controlled) processing while learning a new task (*Wilkins & Rawson, 2011*). The experimental stimuli are presented in Fig. 3.

The picture pattern used for each numerosity was unique; at least two dots differed in location on the screen to the picture of each other numerosity. The practice and the experimental stimuli contained dots 1.5 cm in diameter. Six to eleven dots were used

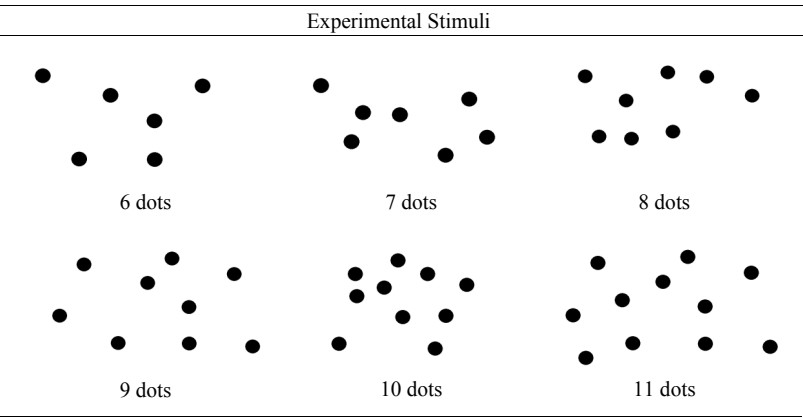

Experimental Stimuli

6 dots 7 dots 8 dots

9 dots 10 dots 11 dots

**Figure 3** **Experimental stimuli used in the 30 experimental blocks.**

to eliminate smaller patterns being 'subitized' which could have masked the observable learning effect over practice (*Green, 1997*). Subitizing results from easier recognition of smaller numbers of items, in turn resulting in more rapid and accurate responding for numbers below six, which would mask any variability in counting reaction times (*Jensen, Reese & Reese, 1950*).

The dot stimuli were presented on a computer monitor. Participants sat with their heads approximately 60 cm from the monitor screen. Each dot stimulus was 12 cm high and 20 cm wide. As a result, each stimulus subtended a vertical visual angle of 11.5° and a horizontal visual angle of 19.6°.

A SuperLab RBx30 series response pad was attached to the computer to collect key responses and reaction time on each trial. Buttons on the response box were arranged in two horizontal rows. The six keys in the first row were labeled, from left to right, 6, 7, 8, 9, 10 and 11, corresponding to the number of dots that were presented in the experimental stimuli. The second row had two keys, both labelled "NEXT".

## Procedure

This project received ethics approval from the Psychology Ethics Sub-Committee of the Edith Cowan University Human Research Ethics Committee prior to commencement (Project # 12590 SHADBOLT). Written consent was obtained from each participant at the beginning of the experiment.

Instructions for each group were presented on the computer screen. These instructions outlined the nature of the counting task, and informed the aware group only that the dot patterns would repeat, specifically drawing their attention to this characteristic of the design. The instructions presented to the aware group were: "The task will require you to watch the computer screen and indicate the number of dots shown on the screen as accurately and quickly as possible. Use the response box provided with the corresponding keys. You should be aware that the dot patterns will be repeated many times in the experiment". The

**Table 1** Post-experiment questions.

**Control group**

1. "Did you notice the repetition of the patterns of each number of dots?" (If response is "No", ask no further questions. If response is "Yes", ask question 2)

2. "Did you find that the repetition of the pattern for each number of dots helpful?" (If response is "No", ask no further questions. If response is "Yes", ask Question 3)

3. "At what point did you find the repetition helpful during the 30 blocks of trials? Can you give an estimate of the block number?"

**Aware group**

1. "Did you find knowing about the repetition of the patterns helpful during the task? (If response is "No", ask no further questions. If response is "Yes", ask Question 2)

2. "At what point did you find the repetition useful during the 30 blocks of trials? Can you give an estimate of the block number?"

instructions presented to the control group did not contain the last instruction sentence presented to the aware group.

All practice and experimental trials had the same structure. At the beginning of each trial a fixation point in the middle of the screen appeared for 500 ms. The dot picture was then presented and remained visible until the participant responded. After a response was made a feedback message (Correct/Incorrect) appeared on the screen for 2 s or until a "NEXT" key was pressed. This was followed by the fixation point for the next trial. All participants completed four practice trials to familiarise them with the response keys. In the experimental trials, each stimulus was presented three times in an 18-trial block. The presentation order of stimuli within a block was random. Participants completed 30 experimental blocks with optional breaks in between each block, resulting in 90 repetitions of each dot pattern.

After the completion of the task a short post-experiment interview was conducted. Questions were asked verbally in order to explore whether the participants recognised the repetition of patterns and when automaticity developed (Table 1). Each session took approximately 40–50 min to complete.

## RESULTS

The data collected in this experiment are freely available at http://osf.io/cnr5z.

### Accuracy

Mean accuracy scores for the 34 participants across all experimental blocks ranged from 89.81% to 99.62%. Minimum accuracy per experimental block ranged between 55.56% and 94.44%.

### Comparison of groups (RT)

RT data was screened prior to analysis. There were only 12 trials over the whole experiment in which RT might be considered slow (>10,000 ms), but these values were not a great deal longer than the vast majority of values (i.e., all but one was less than 15,000 ms, and the largest was less than 20,000 ms), and so were not considered extreme and worth deleting. Furthermore, given the aim of the experiment was to observe automatic performance, we expected some RTs to be very fast, so deleting such trials would be counterproductive. In

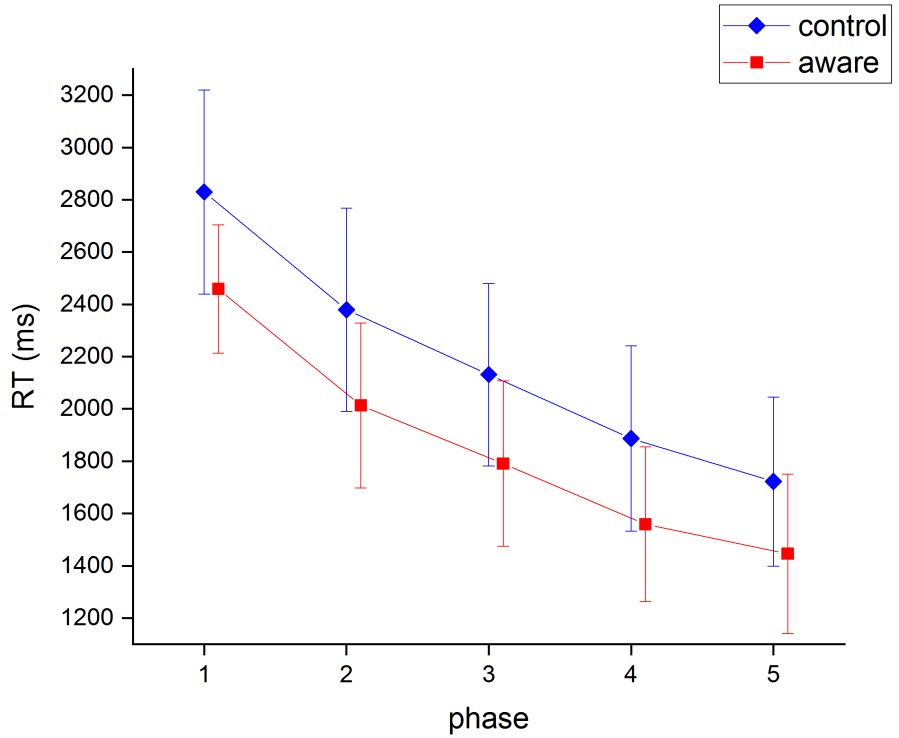

**Figure 4  Mean RT as a function of experimental phase and group.** Error bars are 95% confidence intervals.

the end, though, there were no trials in which a correct response was provided in less than 400 ms.

Data recorded from the 30 experimental blocks were analysed in five phases of 90 trials each. This reduced variability within the data which otherwise would have made clear patterns difficult to detect. Mean RTs were calculated for each of the five phases per person.

A mixed analysis of variance (ANOVA) was conducted in order to examine the mean RTs for the within-subject effect of practice, and the between-subject effect of group. As the assumption of sphericity was violated, the Greenhouse-Geisser adjustment was used. This ANOVA showed there was a significant reduction in RT over the five phases, $F(1.94, 62.15) = 130.68$, $p < .001$, partial $\eta^2 = .803$, but no significant difference between groups, $F(1, 32) = 2.72$, $p = .109$, partial $\eta^2 = .078$. There was no significant interaction effect for groups over practice either, $F(1.942, 62.15) = .27$, $p = .758$, partial $\eta^2 = .008$. Pairwise comparisons showed significant reductions in RT from one phase to the next ($M_{\text{phase 1}} = 2644.28$ ms, $M_{\text{phase 2}} = 2{,}196.21$ ms, $M_{\text{phase 3}} = 1{,}961.31$ ms, $M_{\text{phase 4}} = 1{,}722.94$ ms, $M_{\text{phase 5}} = 1{,}584.25$ ms). These results are depicted in Fig. 4.

Effect size and confidence intervals for the RT time data were examined following the insignificant results of the ANOVAs. The RTs for both groups appeared to decline at a similar rate, with the aware group showing on average faster RTs (Fig. 4). Although RTs appear to be faster for the aware group, Fig. 4 does not indicate any obvious interaction,
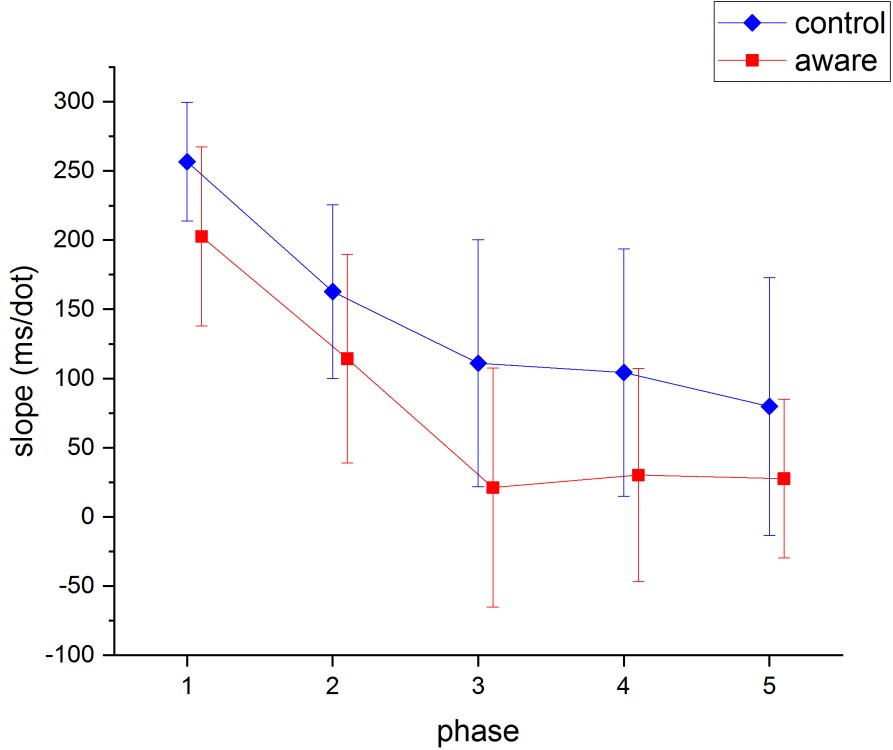

**Figure 5** **Mean slope (RT/dot) values as a function of experimental group and phase.** Error bars are 95% confidence intervals.

and the size of the group effect on performance is small (partial $\eta^2 = .078$). The 95% confidence intervals provide some evidence of a difference in RT between the groups since the intervals of the aware group do not overlap with the mean RT of the control group for phases 1–4 (*Cumming, 2014*).

A further test of the difference between the groups was conducted using Bayesian analysis. A JZS Bayes version of the repeated measures ANOVA (*Love et al., 2015*; *Rouder et al., 2012*) with default prior scales was performed. The Bayes Factor for the group effect was $BF_{10} = 1.01$. This result is equivocal in terms of deciding between the null hypothesis of no difference between the groups, and the alternative hypothesis of there being a difference. Further, the main effects model was preferred to the interaction model by a Bayes factor of 12.68. The data provide evidence against the hypothesis that condition and practice interact in RT.

## Comparison of groups (slope values)

Prior research has established that in the dot counting task there is a clear linear relationship between numerosity (the number of dots in a picture) and RT, at least early in practice (*Lassaline & Logan, 1993*). Due to this well-established relationship, the slope of a linear function relating RT and numerosity was calculated for each phase for each participant. The mean slope values for both groups in all phases are presented in Fig. 5.

A mixed design ANOVA examined the slope values for the within-subject effect of practice and the between-subject effect of group. Again due to violation of the sphericity assumption, the Greenhouse-Geisser adjustment was used. This ANOVA showed a significant reduction in slope values over practice, $F(2.84, 90.85) = 28.62$, $p < .00$, partial $\eta^2 = .472$, no significant difference between groups, $F(1, 32) = 2.29$, $p = .14$, partial $\eta^2 = .067$, and no interaction effect for groups over practice, $F(2.84, 90.85) = .40$, $p = .74$, partial $\eta^2 = .012$. Pairwise comparisons showed a significant decrease in the slope values between the first, second and third blocks only ($M_{\text{phase 1}} = 229.68$ ms/dot, $M_{\text{phase 2}} = 138.55$ ms/dot, $M_{\text{phase 3}} = 66.11$ ms/dot). A JZS Bayes version of the repeated measures ANOVA (*Love et al., 2015*; *Rouder et al., 2012*) with default prior scales was performed. The Bayes Factor for the group effect was $BF_{10} = 0.85$, again providing equivocal support for there being no difference between the groups. Further, the main effects model was preferred to the interaction model by a Bayes factor of 11.95. The data provide evidence against the hypothesis that condition and practice interact in the slope values.

## Automaticity attainment

The point at which participants reached automaticity was compared between groups. On the basis of the results reported by *Lassaline & Logan (1993)*, reaching automaticity was defined as the block in which the slope of the RT by numerosity function reached 100 ms/dot or less. Average slope values significantly decreased over practice until approximately reaching 100 ms/dot (phase 3) after which they did not significantly decrease with more practice (see Fig. 5). A similar pattern of slope value change was reported by Lassaline and Logan, with no significant reductions after approximately 100 ms/dot.

Participants in the control and aware groups were compared using the 100 ms/dot cut-off value. A participant was considered to have attained automaticity in the experimental block in which they reached this value. All but two participants (one in the aware group and one in the control group) reached a slope value of 100 ms/dot or less, thus the data for the two subjects who did not reach the criterion were excluded from this particular analysis. There was no difference found in the point at which automaticity was attained between the two groups ($M_{\text{aware}} = 7.25$ experimental blocks of practice, $M_{\text{control}} = 7.25$ experimental blocks of practice), as depicted in Fig. 6.

## Participants' awareness

A correlational analysis was conducted in order to review the relationship between when, in the post-experimental interview questions, participants said they learned and used the patterns and when they reached automaticity, according to the 100 ms/dot criterion. All participants in the control group indicated that they were aware of the repetition and found it helpful to complete the task. Further, all participants in the aware group indicated that they felt being told about the repetition before the task was helpful. An analysis then compared the point at which subjects estimated that they used the repetition of items in their performance of the task with the point at which their data indicated that they had attained automaticity, but only with the data of the 32 participants who reached automaticity. Pearson correlation coefficients indicated that there was no correlation
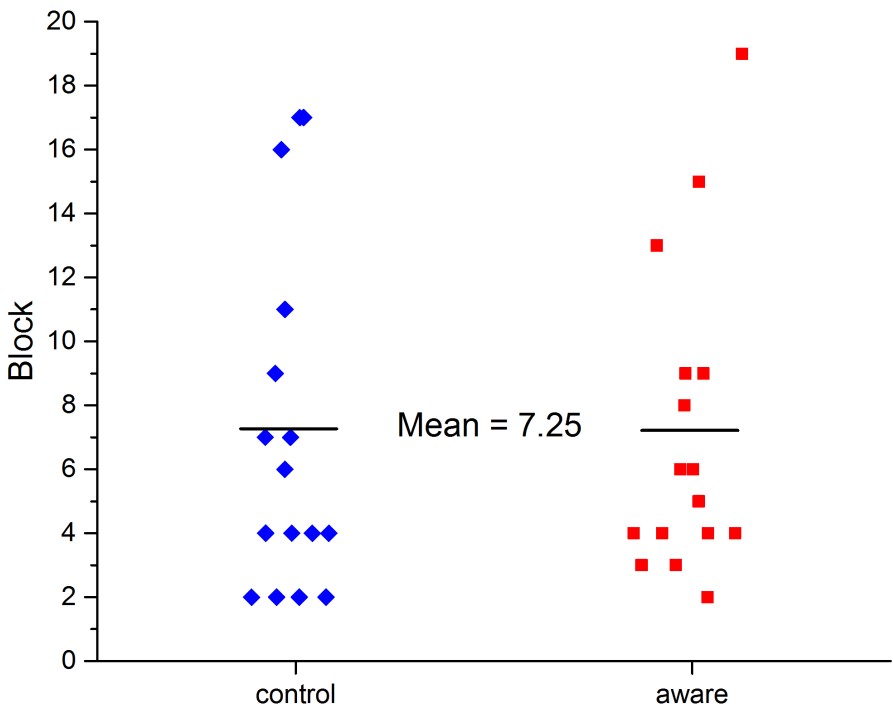

**Figure 6** **The experimental block number in which participants reached automaticity (slope ≤ 100 ms/dot).** The horizontal lines indicate the average block in which participants in each group reached automaticity.

between these two points for the aware group $r = .016$, $p = .95$, or the control group, $r = .276$, $p = .30$. An independent samples $t$-test compared the two groups on the point at which subjects estimated that they used the repetition of items in their performance of the task ($M_{aware} = 8.82$ blocks, $SD = 7.65$; $M_{control} = 11.76$ blocks, $SD = 5.86$). This difference was not statistically significant, $t(32) = 1.26$, $p = .217$. A Bayes independent samples $t$-test provided equivocal support for the null hypothesis ($BF_{10} = 1.63$).

## DISCUSSION

The results of this experiment provide no evidence that being aware of the repetition of items in the dot counting task facilitates the development of automaticity in the task. Two results support this conclusion. The first is that there was no discernible interaction between the effects of group and practice. Although there was some suggestion in the data that the aware group was generally faster than the control group, this is likely to just reflect individual differences in dot counting speed, which persisted at a constant level throughout the experiment. The small speed difference between the groups was obvious from the beginning of the experiment, when knowing about the repetition of the items could not arguably be of much advantage. Importantly, though, being aware of the repetition of the dot stimuli did not result in the aware group increasing their speed advantage as practice continued.

The second result that supports the conclusion that awareness of the repetition of the items did not facilitate the attainment of automaticity was that there was no difference between the groups in the reduction of slopes with practice. Most importantly, there was no interaction between the group and practice variables, suggesting that both groups transitioned to automaticity at the same rate. Further, the average point at which participants reached the point of automaticity was similar for both groups. Thus, there seems to be no advantage of being aware of the item repetition before the task commenced.

The results also showed that regardless of condition, the participants did not seem aware of when automaticity had been attained. All participants reported recognising the repetition in the stimuli, but were apparently unable to accurately report when they felt this helped them perform the task. There was no correlation between when they felt the item repetition helped them perform the task, and when the data showed that automaticity was reached.

It is worth noting, before the results are interpreted further, that the conclusions rely on accepting null hypotheses, and that the lack of evidence for rejecting these hypotheses could simply reflect a lack of power. Nonetheless, there was sufficient power to detect some effects in this data, so a lack of power would indicate that what look like non-effects in this data set were actually very small effects, and so unlikely to be important.

The instance theory (*Logan, 1988*; *Logan, 1990*; *Logan & Klapp, 1991*) of automaticity provided the context and grounding from which this study was conducted. The results of this study support the nature of automaticity described by Logan in the instance theory. They also indicate that participants are not aware of when they develop automaticity, and that automatic responding is not subject to the influence of awareness of extra-trial factors. The expected advantage of knowing about the repetition of stimuli resulting in faster reductions in reaction time did not occur. Given there was no difference in performance improvement between the aware and control groups, being aware of the repetition did not provide any advantage. These results are consistent with instance theory as the theory defines automatic processing as requiring no attention, making performance seemingly effortless, fast and unavailable to conscious influence such as directed attention to extra-trial information. The results of this study supported the theory that automatic responses are based on memory retrieval alone once the task is practiced enough for each stimulus to elicit a response based on recognition. More specifically, the results support a memory-based explanation of automaticity where each processing episode is stored separately, and performance speed-up is a statistical effect of storing many instances.

The results of the experiment indicate that the time at which people reported responding on the basis of memory retrieval did not correlate with the point at which their reaction times indicated they had reached automaticity. It is interesting to note that there was no significant difference between the groups regarding when they felt they reached this point. However the aware group reported that knowing about the repetition in the pre-experimental instructions helped them perform the task quicker. These results indicate that being told about the repetition in the task enabled participants to think they performed better than if they had not been told, even if it made no difference to their performance or awareness of when automaticity developed. It appears, then, that people are not aware

of when the change from controlled to automatic processing occurs. One possible reason for this may be that, as the focus of participants was to respond as accurately and rapidly as possible, they did not attend to the way in which they completed the task, only to completing it. Post-experiment questioning then may be asking participants to try to retrieve information that was no longer available to report.

It is possible that participants in the aware condition just ignored the additional sentence in the instructions regarding the repetition of items. If this were the case, this would provide a trivial explanation of why there were no differences between the two groups. There is some indication, however, that this was not the case. For instance, if the aware participants did ignore this sentence, it would be expected that at least some would have remarked about this, or at least indicated that this information was not helpful. The current data cannot completely rule out the "ignoring" hypothesis. Some additional test would be required.

## CONCLUSION

The purpose of this study was to investigate whether bringing awareness to extra-trial information can improve the speed at which automaticity developed. Given the evidence that where attention is directed during the performance of a task can determine whether or not automaticity is developed, it was hypothesised that prior awareness of the repetition of stimuli in the dot counting task could aid the speed of developing automatic processing. The results demonstrated that this was not the case. The current findings suggest no advantage resulted from being aware of the repetitious nature of the task before learning began, and that the rate of developing automatic processing was constant regardless of being aware or not of the broader context. Development of automaticity being immune to the influence of extra-trial information is consistent with instance theory, which postulates automatic processing being an unconscious process requiring no attention.

In conclusion, this study is consistent with Logan's research and his instance theory explanation of automaticity. It appears that the speed of developing automatic processing in a numerosity task is not influenced by pre-awareness of the repetition of stimuli.

### Funding
The authors received no funding for this work.

### Competing Interests
The authors declare there are no competing interests.

### Author Contributions
- Craig P. Speelman conceived and designed the experiments, analyzed the data, wrote the paper, prepared figures and/or tables, reviewed drafts of the paper.
- Emma Shadbolt performed the experiments, analyzed the data, wrote the paper, prepared figures and/or tables, reviewed drafts of the paper.

## Human Ethics

The following information was supplied relating to ethical approvals (i.e., approving body and any reference numbers):

The Psychology Ethics Sub-Committee of the Edith Cowan University Human Research Ethics Committee granted Ethical approval to carry out the study within its facilities (Project # 12590 SHADBOLT).

## Data Availability

The raw data has been provided as a Supplemental File.

## Supplemental Information

Supplemental information for this article can be found online at http://dx.doi.org/10.7717/peerj.4329#supplemental-information.

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
