# Peer review of "The role of awareness of repetition during the development of automaticity in a dot-counting task"

_PeerJ, doi:10.7717/peerj.4329_

## Round 0.1 · original submission · Major Revisions

Dear Craig,

Thanks again for your submission and patience with the review process.

Many thanks to the two reviewers who have assisted with this paper. With their complementary backgrounds, both have praised the clarity of the work and raised points for its further contextualisation within the current literature. I feel that the reviewers' comments are sufficiently clear that I won't reiterate them at this point but if feels that the question of power is particularly poignant, particularly regarding the inconclusive level of the Bayes factors. Although none of the feedback reflects fatal flaws, given the work involved in addressing the comments, I have recommend Major Revisions.

I have a few comments of my own that I hope will be helpful:
Please consider reporting the size of stimuli in visual angle rather than cm (lines 184-185). This will make replication more straight forward. Participant viewing distance would be important to include to clarify the visual angle as well.

Was any screening of reaction time values conducted? (e.g. excluding particularly faster < 250 sec values) If not, probably a good idea to confirm this in the method at some point.

Just on the screening, considering the often skewed distribution of reaction time data, I’m used to seeing the median used to describe individual performance so that group summary would be the group mean of the median reaction times. This might not be appropriate for the current data set but I just wanted to flag it in case it adds sensitivity to the data.

I find the column scatter plot (Figure 6) particularly informative - really nice to see all the data points. It’d be nice to see these included on the other figures - x-axis offsetting to the left and right would clearly delineate the groups but also make the overlap of the error bars easier to see.

I hope the feedback is useful and that a revision is within your resources. Please feel free to get in touch if I can be of assistance.

Best wishes,
Nic

·

Basic reporting

The paper is well written, and the description of the design and presentation of the results is clear. The review of the literature is relevant and clear, but does not include some literature that I think is relevant. In particular, the manipulation and assessment of awareness in learning has been a fraught topic for decades, and while it is true that these questions have not been studied in tasks closely resembling the present one, the large literature on implicit sequence learning that has developed over the last 20 years or so contains much relevant consideration of both theoretical and methodological issues. I think that citing this literature would be useful, and might help to the authors’ choice of manipulation and assessment.

Experimental design

The research question is clearly identified, and the experimental design and procedure are straightforward. However, the logic linking the reviewed theory to the design seems to have a gap. My understanding of Logan’s theory is that each instance is an independent memory representation. Therefore, it’s not clear that that the theory attention to, or awareness of, repetition per se should affect the rate of learning. It seems to me that the manuscript does not clearly connect the statements in lines 120-139, which focus on attention to characteristics of the stimulus and response (i.e., pattern and numerosity) to the manipulation, which focuses on the relation between rather than within trials. In fact, on my reading, Logan’s theory would not predict an effect of this particular manipulation of awareness (because each instance is a separate memory representation), though it may predict an effect of awareness of the relation between pattern and numerosity within instances. In this sense, the null results may support the theory in a more specific way than described in the manuscript. On the other hand, the authors may have an argument, left implicit in the present manuscript, that links awareness of repetition with attention to the task elements critical for forming an instance representation.

It would be helpful to better defend the specific manipulation of awareness in terms of the hypothesized effect on the development of automaticity, if that is possible, and to explicitly contrast the content of this awareness with other possible contents that might have different theoretical relations to the phenomena.

Validity of the findings

The data provide a robust replication of well-established effects important to instance theory (and to other theories of skill acquisition). As the authors note, though, their important conclusions depend on accepting the null hypothesis concerning the effect of awareness of repetition. This may be theoretically significant in the present case, but no power analysis is presented. It is true, as the authors say, that there was sufficient power to detect some effects – overall speedup and reduction in slope – but these are known to be very large effects. However, I did not find this a convincing argument that there was sufficient power to detect important effects. The number of participants is quite small for a between-subjects design intended to detect an effect of unknown size. Moreover, as the authors also note, it is possible that awareness was not in fact successfully manipulated – the design included no manipulation check. In fact, it’s not clear how participants would have used awareness of repetition on individual trials – by hypothesis, the retrieval of instances is automatic in the sense that the appearance of the stimulus is supposed to trigger recognition and thus instance retrieval.

I did think that the authors were admirably clear in pointing out the limits of their results and conclusion, and in considering alternative hypotheses.

Additional comments

I’m sympathetic to the aims of the study – examining the effects of awareness on a type of learning that has been important to theory in the field. I wish, however, that the manipulation were more clearly linked to the theory, and the robustness of the null finding were easier to evaluate.

·

Basic reporting

The English is clear, unambiguous, and professional.

The article structure is professional, and easily understood.

The article is self-contained.

The term "automatic" is not consistently defined across studies (see Logan, 1988, and more recently Reynolds & Besner, 2006). I think it's important that researchers in the area are explicit about what they mean by the term, and which features of automaticity they are concerned with. I believe a discussion of this problem is warranted.

Data is shared, but only in aggregate form making it impossible to replicate the analysis from scratch, or conduct alternative analyses with different decisions around outlier removal etc. I would prefer to see the original trial-level data included as well (anonymized, of course).

General Statement on Open Science
I think that open science principles are critical to improving research, and have adopted a policy of promoting them in my reviews. To this end, I would invite the authors to provide the data, materials, and analysis files/scripts in a public repository such as http://osf.io so that others can verify and re-examine claims, use the results to inform their own study designs, and ensure a more complete record of experiments to counteract publication bias in meta-analyses. Alternately, manuscripts should include a clear statement justifying the decision not to provide these materials. (See opennessinitiative.org for more on open science practices)

Experimental design

I leave it to the editor to determine if the article fits within the scope of the journal, though I see no reason why it shouldn't, given other articles that have appeared here.

The research question could be clarified with a clearer statement of how automaticity is defined and operationalized here.

The design is an extension of a paradigm used by Lassaline & Logan (1993). The authors test more subjects (17 per condition rather than 4) but with far fewer trials (540 vs. 6000+) per subject, and conducted in a single session rather than across 13 sessions. I think a discussion of these differences is warranted (it's worth noting that Speelman and Townsend (2015) used a similar design, and observed the same pattern of a reduction in the slopes that Lassaline and Logan observed).

Validity of the findings

This is the area of greatest concern to me.

1. If I understood correctly, the index of "time to automaticity" is the first block of 18 trials in which the subject achieves a slope of 100ms or less. However, it's quite clear from the data that in many cases the participants do not *remain* below 100ms from this point on, which raises the question of whether the task can really be considered to have become automated at that point, or if perhaps some index that takes into account the stability of that slope going forward is needed. The slope for each block is, after all, based on a maximum of 18 trials (3 for each number of dots). This is likely to be highly volatile.

2. Relatedly, the authors have adopted this 100ms/dot criterion as the index of automaticity. However, many of the subjects seem not to have achieved that measure consistently during Phase 5 (given the error bars there). How do the authors reconcile these with their conclusion that automaticity is achieved? Perhaps Speelman and Townsend's findings that not all subjects achieve automaticity in this task would be worth discussing here.

On reporting and interpreting Bayes factors:
3. I applaud the authors for using both Bayesian and traditional Null Hypothesis techniques to address the question, however the conclusions they draw from the BFs are misleading. Bayes factors around 1 are equivocal: providing little evidence in either direction. Statements such as "A similar conclusion is suggested by a a Bayes .... (Bayes Factor (.05) = 1.63)" (line 301-302) are thus misleading. The Bayes factor of 1.63 suggests no ability to draw conclusions of any kind.

4. The key hypothesis here was that subjects in the aware condition might achieve automaticity faster than subjects in the control condition. This would be indicated by the presence of a group x phase interaction for the slopes such that the aware group's slopes dropped faster than the control group. The ANOVA analysis considers this interaction directly, but the Bayes Factors are reported only for the main effect of group. Why are Bayes Factors for the critical interaction not considered?

5. I'm unfamiliar with the notation "Bayes Factor (.05)". What is the "(.05)" here? Similarly, no indication is given of which model the Bayes factor favours. BF=1.63 (line 302) may indicate slightly more evidence for the null, or for the alternative. (Based on the t statistic, I think the BF likely favours the null here.) This ambiguity needs clarification throughout.

6. The BFs all being around 1 also speaks to the statements in lines 326-330. BFs in that range should be taken to imply that more data is needed to discriminate the two hypothetical models (null, vs. alternative). They certainly can not be taken to support the view that since some effects are significant, there likely isn't a power problem. However,
I reiterate that the BFs reported do not appear to directly address the critical interaction.
(It's also worth noting that the significant effects are all main effects of phase, while the critical, non-significant results are interactions - ANOVA is more powerful for main effects, so the presence of those effects does not offer much defence against power concerns.)

Additional comments

The article raises an interesting question (can awareness of relevant features of stimuli speed the process of moving from "slow and effortful" to "fast and automatic"). The design of the study is a simple extension of one used before, however I wonder at whether the design is well-enough powered to conclude there is no effect. The use of Bayesian analysis is great, but as applied here does not help to address the most important feature of the data.

There is a significant benefit to the decision to use Bayes Factors. If the BFs for the interaction are equivocal as they are for the main effects of group, the the authors are welcome to simply collect more data until the Bayes Factor is able to discriminate the base model (main effects of Phase and Group) from a model with the addition of the interaction.

I also worry about the way that "achieving automaticity" is operationalized here since subjects seem to go from the "automatic" 100ms/dot to "effortful" 101+ms/dot from block to block (though it's hard to be sure how much this occurs, since the data provided are aggregated into phases of 6 blocks.) This suggests to me that there is a high level of volatility in the measure as an index of automaticity.

I'm not sure how difficult it will be to address my concerns, but it seems to me that it isn't impossible so I am currently recommending "Major Revisions."

---

## Round 0.2 · accepted · Accept

Dear Craig and Emma,

Thanks for the updates to the manuscript. As you will see, we have received follow-up reviews from the two original reviewers, both indicating that they are happy with the updates and that the article is suitable for publication. Congratulations.

I would recommend the Dienes (2011) paper on Bayes suggested by the second reviewer. It feels that a Bayesian stopping rule may be an efficient strategy to employ in future studies on this subject.

Best wishes for the future,
Nic

·

Basic reporting

The paper now cites work on the role of attention in the literature on implicit learning, as I suggested.

Experimental design

The authors have clarified the rationale for their predictions and how they follow from instance theory. Making this explicit is a major improvement.

Validity of the findings

The authors have provided a more appropriate and robust statistical approach to supporting the robustness of the finding of no interaction.

Additional comments

The authors have responded effectively to my earlier concerns, so I have limited this re-review to noting that. I think the revision greatly clarifies the contribution of this work.

·

Basic reporting

no comment

Experimental design

no comment

Validity of the findings

no comment

Additional comments

see attached